# Associations between Smoking and Alcohol and Follicular Lymphoma Incidence and Survival: A Family-Based Case-Control Study in Australia

**DOI:** 10.3390/cancers14112710

**Published:** 2022-05-30

**Authors:** Michael K. Odutola, Marina T. van Leeuwen, Jennifer Turner, Fiona Bruinsma, John F. Seymour, Henry M. Prince, Samuel T. Milliken, Judith Trotman, Emma Verner, Campbell Tiley, Fernando Roncolato, Craig R. Underhill, Stephen S. Opat, Michael Harvey, Mark Hertzberg, Geza Benke, Graham G. Giles, Claire M. Vajdic

**Affiliations:** 1Centre for Big Data Research in Health, University of New South Wales, Sydney 2052, Australia; m.odutola@unsw.edu.au (M.K.O.); m.vanleeuwen@unsw.edu.au (M.T.v.L.); 2Department of Anatomical Pathology, Douglass Hanly Moir Pathology, Macquarie Park 2113, Australia; jtur8838@bigpond.net.au; 3Department of Clinical Medicine, Faculty of Medicine, Health and Human Science, Macquarie University, North Ryde 2109, Australia; 4Cancer Epidemiology Division, Cancer Council Victoria, Melbourne 3004, Australia; fiona.bruinsma@cancervic.org.au (F.B.); graham.giles@cancervic.org.au (G.G.G.); 5Centre for Epidemiology and Biostatistics, Melbourne School of Population and Global Health, University of Melbourne, Parkville 3010, Australia; 6Royal Melbourne Hospital, Melbourne 3052, Australia; john.seymour@petermac.org; 7Peter MacCallum Cancer Centre, University of Melbourne, Parkville 3010, Australia; miles.prince@petermac.org; 8Epworth Healthcare, Richmond 3121, Australia; 9St. Vincent’s Hospital, Sydney 2010, Australia; smilliken@stvincents.com.au; 10University of New South Wales, Sydney 2052, Australia; fernando.roncolato@health.nsw.gov.au (F.R.); mhertzberg10@gmail.com (M.H.); 11Concord Repatriation General Hospital, Concord 2139, Australia; judith.trotman@health.nsw.gov.au (J.T.); emma.verner@health.nsw.gov.au (E.V.); 12Faculty of Medicine and Health, University of Sydney, Concord 2139, Australia; 13Gosford Hospital, Gosford 2250, Australia; campbell.tiley@health.nsw.gov.au; 14School of Medicine and Public Health, The University of Newcastle, Newcastle 2308, Australia; 15St. George Hospital, Kogarah 2217, Australia; 16Rural Medical School, Albury 2640, Australia; craig.underhill@bordermedonc.com.au; 17Border Medical Oncology Research Unit, Albury 2640, Australia; 18Clinical Haematology, Monash Health and Monash University, Clayton 3168, Australia; stephen.opat@monashhealth.org; 19Liverpool Hospital, Liverpool 2170, Australia; michael.harvey@sswahs.nsw.gov.au; 20Western Sydney University, Sydney 2000, Australia; 21Department of Haematology, Prince of Wales Hospital, Sydney 2031, Australia; 22School of Public Health and Preventive Medicine, Monash University, Melbourne 3004, Australia; geza.benke@monash.edu; 23Precision Medicine, School of Clinical Sciences at Monash Health, Monash University, Clayton 3168, Australia; 24Kirby Institute, University of New South Wales, Sydney 2052, Australia

**Keywords:** follicular lymphoma, smoking, alcohol, incidence, survival

## Abstract

**Simple Summary:**

Previous studies on the relationship between smoking and follicular lymphoma (FL) incidence and survival are inconsistent, while the evidence regarding alcohol intake appears to support an inverse association. In this population-based family case-control study, we observed a positive association between multiple measures of personal smoking history and increased risk of FL, including evidence of a dose-response. We also observed an association between personal smoking and poorer overall survival after FL diagnosis and an indication that personal smoking may be associated with poorer FL-specific survival. Additionally, among non-smokers, we found increased FL risk for those exposed indoors to more than two smokers during their childhood. In contrast, we observed no evidence of an association between recent alcohol consumption and risk of FL, overall survival, or FL-specific survival. Our findings further strengthen the evidence for ongoing multi-faceted tobacco control activities to reduce FL incidence and improve patient outcomes.

**Abstract:**

The association between smoking and alcohol consumption and follicular lymphoma (FL) incidence and clinical outcome is uncertain. We conducted a population-based family case-control study (709 cases: 490 controls) in Australia. We assessed lifetime history of smoking and recent alcohol consumption and followed-up cases (median = 83 months). We examined associations with FL risk using unconditional logistic regression and with all-cause and FL-specific mortality of cases using Cox regression. FL risk was associated with ever smoking (OR = 1.38, 95%CI = 1.08–1.74), former smoking (OR = 1.36, 95%CI = 1.05–1.77), smoking initiation before age 17 (OR = 1.47, 95%CI = 1.06–2.05), the highest categories of cigarettes smoked per day (OR = 1.44, 95%CI = 1.04–2.01), smoking duration (OR = 1.53, 95%CI = 1.07–2.18) and pack-years (OR = 1.56, 95%CI = 1.10–2.22). For never smokers, FL risk increased for those exposed indoors to >2 smokers during childhood (OR = 1.84, 95%CI = 1.11–3.04). For cases, current smoking and the highest categories of smoking duration and lifetime cigarette exposure were associated with elevated all-cause mortality. The hazard ratio for current smoking and FL-specific mortality was 2.97 (95%CI = 0.91–9.72). We found no association between recent alcohol consumption and FL risk, all-cause or FL-specific mortality. Our study showed consistent evidence of an association between smoking and increased FL risk and possibly also FL-specific mortality. Strengthening anti-smoking policies and interventions may reduce the population burden of FL.

## 1. Introduction

Follicular lymphoma (FL) is a common subtype of non-Hodgkin Lymphoma (NHL) that arises from follicle center B-cells [1]. Most FLs display an indolent clinical behaviour, although 5.2–8.7% transform over 10 years to diffuse large B-cell lymphoma (DLBCL), an aggressive NHL subtype [2]. Epidemiological evidence suggests heterogeneity in the etiology of NHL subtypes, necessitating studies focused on FL [3]. t(14;18) chromosomal translocation is present in 80–89% of FL cases [4,5] and is considered the initiating event in lymphomagenesis. This translocation is also found in the lymphocytes of apparently healthy individuals, with increased frequency with increasing age [6]. Smoking similarly increases t(14;18) frequency [7]. 

Results from a recent meta-analysis of personal smoking and FL risk were inconsistent and non-significant, with mixed findings in cohort and case-control studies for former and current smoking [8]. The only study to examine passive smoking and FL risk in never smokers found excess risk for those exposed during childhood and adulthood, and significant trends in risk with increasing duration and intensity of exposure [9]. A meta-analysis found consistent evidence of reduced FL risk for current alcohol intake in cohort studies, no association in case-control studies, and no evidence of an association with former intake or the type of alcohol consumed [8].

Data on the association between smoking and alcohol and clinical outcomes of FL are limited. A meta-analysis reported higher risk of all-cause mortality with higher number of cigarettes smoked per day, longer duration and higher pack-years of smoking [10]. Evidence to-date suggests no association with earlier initiation age, intensity, duration or lifetime consumption of alcohol or specific alcohol type [11,12,13,14]. The only study to examine alcohol consumption and FL-specific mortality [11] also found no association. 

To further investigate these associations, we conducted a population-based, family case-control study examining the relationship between smoking and alcohol and FL risk and survival after diagnosis in Australia.

## 2. Materials and Methods

### 2.1. Study Sample

Cases were diagnosed between 2011 and 2016, aged 20–74 years and resident in New South Wales (NSW) or Victoria, the two most populous states in Australia [15]. Cases were eligible if they had histopathologically confirmed FL (including composite FL and DLBCL), no prior history of haematopoietic malignancy and provided informed consent. We identified 1791 cases: 125 cases via participating clinics and 1666 via the NSW or Victorian Cancer Registry. All cases had a central review of diagnostic histopathology reports and ancillary tests, including flow cytometry. A total of 213 cases, enriched for those with low confidence in diagnosis on the basis of pathology report review, underwent diagnostic slide review by an expert histopathologist (JT) based on a previously published methodology [16]; this review identified 13 ineligible cases where the pathological diagnosis was not confirmed. Of the 1778 eligible and contactable cases, 733 declined and 1045 (58.8%) consented to be approached by the study coordinating centre. Of those approached by the study, 77 cases could not be reached, 770 (79.5%) were enrolled and 198 (20.5%) declined. Of those enrolled, 709 cases (92.1%) completed the smoking and alcohol questionnaires (Appendix A).

We enrolled related and unrelated controls aged 20–74 years with no history of haematopoietic malignancy who provided informed consent. Case participants provided consent for family members to be invited to participate in the study. When a case had multiple siblings, and consented to all of them being approached, the sibling of the same sex and closest age was approached first. Where cases had no siblings or no living, eligible or consented siblings, they nominated their spouse or partner. Of those approached, 65 controls could not be reached for a response, 517 (80.0%) consented to participate and 130 (20.0%) declined. The participation rates for unrelated and related controls were nearly identical (79.8% and 80.0%, respectively). Of those enrolled, 490 controls (94.8%) completed the smoking and alcohol questionnaires (Appendix A). 

Ethical approval for the study was obtained from the NSW Population and Health Services Research Ethics Committee (2011/07/337).

### 2.2. Exposure Variables

Information on smoking status and alcohol intake was collected using a structured questionnaire, and participants completed a lifetime residence and work calendar to aid recall [17,18]. They reported their history of regular personal cigarette smoking and also passive exposure to cigarette smoke. In the questionnaire, regular smoking was defined as smoking at least 7 cigarettes a week, on average, for at least 1 year. Information collected on personal smoking included age initiated, average number of cigarettes smoked per day (frequency), years smoked (duration) and age quit. 

Passive smoking was defined in the questionnaire as indoor exposure to cigarette smoke at least 4 days/week for at least a year during childhood (<18 years) or adulthood (≥18 years), including social venues. Participants were asked to report the number of years they were exposed to childhood passive smoking from four different sources: (a) mother, (b) father, (c) brothers and sisters, (d) other people (e.g., friends, other relatives such as grandparents and work colleagues). Exposure from each source was summed to obtain the total cumulative exposure. For example, the total cumulative exposure will be 21 years if: mother smoked in participant’s presence during childhood for 5 years, father for 5 years, brothers and sisters for 5 years, friends for 3 years, grandparents for 2 years and work colleagues 1 year. Participants reported the number of smokers they were exposed to (intensity) in childhood or adulthood (1, 2, 3, 4, 5, 6, >6), and the duration in childhood (1–2, 3–5, 6–8, 9–11, 12–14, 15–17, 18–20, >20 years) and adulthood (1–2, 3–5, 6–10, 11–15, 16–20, 21–30, 31–40, 41–50, >50 years). 

For participants who reported consuming any alcohol in the twelve months prior to enrolment, we collected information on the frequency of intake of any alcohol, and separately, the frequency and quantity of beer, wine and spirits consumed. 

### 2.3. Case Clinical and Outcome Data

Cases’ treating clinician recorded the stage of disease (Ann Arbor criteria; I–IV), serum levels of lactate dehydrogenase (≤ or >institutional normal range), haemoglobin (<12 g/dL or ≥12 g/dL), number of areas of lymph node involvement (<5 or ≥5), β2-microglobulin (≤ or >normal range), largest nodal diameter (≤6 cm or >6 cm) and bone marrow involvement by lymphoma (no, yes, unknown) to allow the calculation of the Follicular Lymphoma International Prognostic Index (FLIPI/FLIPI-2) [19,20]. Clinicians also provided the date and type of first-line treatment (none, radiotherapy and/or chemotherapy). Histologic grade (1-3B) was based on the slide review or extracted from local pathology reports. 

We ascertained deaths to 5 November 2020 through probabilistic record linkage to the National Death Index by the Australian Institute of Health and Welfare; cause of death data are released two years after the date of death. 

### 2.4. Statistical Analysis

#### 2.4.1. FL Incidence

We classified ever smokers as former smokers if they had quit smoking more than 24 months prior to FL diagnosis (enrolment for controls), otherwise they were considered current smokers. We calculated pack-years smoked by multiplying the average number of cigarettes smoked/day (divided by 20) by the duration. Continuous variables, including the frequency, duration and pack-years of smoking, were divided into tertiles based on their distribution in exposed controls. We explored the association between smoking and FL risk using two reference categories: never smokers, with or without passive smoking exposure.

We restricted analyses of passive smoking to never smokers to exclude residual confounding by personal smoking [9,21,22]. We classified participants as having no passive smoking exposure or exposure in childhood only, adulthood only, both childhood and adulthood, and exposure as an adult in social venue settings. We used the distribution of exposed controls to identify categories for intensity and duration of passive smoking.

For each alcohol type, we calculated the average daily alcohol intake by multiplying the volume of alcohol consumed (mL) by the daily equivalent frequency of consumption. We then obtained the grams of ethanol consumed per day by multiplying the average daily alcohol intake by the specific gravity and standard grams of ethanol per 100 mL, for each alcohol type [23]. These values were summed to obtain the overall daily ethanol intake (g/day). The reference category for the alcohol analyses was non-drinkers in the 12 months prior to enrolment. 

In our primary analyses, we examined associations using unconditional logistic regression models and estimated odds ratios (ORs) with 95% confidence intervals (CI). We used the maximum likelihood method and applied the *vce* (*cluster clustvar*) option in the model to account for clustering within sibships [24,25]. We tested the linearity assumption for continuous variables. We reviewed the literature [8] and generated directed acyclic graphs using DAGitty to guide decisions for the inclusion of confounders in our multivariable models [26,27]. All models were adjusted by the study design factors: age (years), sex (male, female), ethnicity (Caucasian, other) and state (NSW, Victoria). We further adjusted for frequency of alcohol intake (never, <once, once, >once per week) in the smoking multivariable model, smoking status (never, former, current) in the alcohol multivariable model, and other types of alcohol in analyses focusing on alcohol type (Appendix A) [28,29]. We performed a sensitivity analysis, excluding cases with composite FL/DLBCL or grade 3B histology. We also performed sensitivity analyses stratifying by control source (related, unrelated); we excluded cases without sibling controls in the related-controls conditional logistic regression model, and we included all cases in the unrelated-controls unconditional logistic model [30,31].

#### 2.4.2. FL Mortality

For survival after FL diagnosis, follow-up began at the date of diagnosis and ended at death or 5 November 2020, whichever came first. FL-specific survival was defined as death due to FL; all other deaths were censored. We used Cox proportional hazard regression models to estimate the hazard ratios (HRs) with 95%CI for all-cause and FL-specific mortality associated with smoking and alcohol intake. We adjusted for age, sex, ethnicity and state in the basic model. There was no additional adjustment when examining smoking, while we further adjusted for smoking when examining alcohol (Appendix A). Sensitivity analyses were performed: with adjustment for stage (1–2, 3–4) and initial treatment (none, chemotherapy/radiotherapy); excluding cases with composite FL/DLBCL or grade 3B histology. The Cox proportional hazards assumption was assessed for each variable and no violations were observed.

We performed multiple imputation by chained equations for all analyses under the assumption that missing values were missing at random [32]. Where appropriate, we tested the linear trend of the associations with categorical variables by fitting the median value corresponding to each category and modelling this as a continuous variable. 

All statistical analyses were performed using STATA version 15.0 (STATA Corp., College Station, TX, USA). All statistical tests were two-sided and *p* < 0.05 was considered statistically significant.

## 3. Results

Table 1 shows the characteristics of study participants. The median age was 60.8 (interquartile range (IQR) 52.5–67.1) years for cases, 59.3 (IQR 51.4–65.0) years for related controls and 62.6 (53.9–68.3) years for unrelated controls. Approximately 52% of cases and 41% of controls were men, and most (94%) were Caucasian. For cases, 6.6% were composite FL/DLBCL and 6.2% were grade 3B. There were few current smokers (6.9% controls), whereas any exposure to passive smoke in never smokers (65.2% controls) and current alcohol consumption (88.8% controls) was common. Data on smoking and alcohol intake were missing for 3.0% (36 cases, 24 controls) and 0.5% (7 cases, 2 controls) of participants, respectively.

### 3.1. FL Incidence

Results of the basic and multivariable models for smoking were similar (data not shown). Smoking was positively associated with FL risk and the excess risk was similar forever smokers (OR = 1.38, 95%CI 1.08–1.74) and former smokers (OR = 1.36, 95%CI 1.05–1.77; Table 2). Current smoking was not associated with an elevated FL risk (OR = 1.43, 95%CI 0.92–2.20). We observed a 1.47-fold significant excess risk with smoking initiation before the age of 17 years, but no dose-response with age at initiation. We found no association between FL risk and years since quitting smoking. Similar excess risk was observed for the highest categories of cigarettes smoked per day (OR = 1.44, 95%CI 1.04–2.01), duration (OR = 1.53, 95%CI 1.07–2.18) and pack-years of smoking (OR = 1.56, 95%CI 1.10–2.22); trends in risk for increasing duration and pack-years smoking were statistically significant. Overall, the associations with smoking characteristics remained significant and were modestly yet consistently strengthened when the reference group excluded passive smokers. 

We observed similar positive associations with smoking in models including all cases (*n* = 709) and only unrelated-controls (*n* = 118; Appendix A). FL risk was positively associated with all smoking variables except current smoking. Compared to models with all controls, the point estimates were higher (1.66–2.76), and again slightly strengthened when passive smokers were excluded from the reference category. We found no association between FL risk and smoking in models including only related cases (*n* = 242) and controls (*n* = 303; Appendix A). The highest point estimate (1.68) was observed for current smoking and while most ORs were greater than 1, the confidence intervals were generally wide, and the point estimates were not strengthened when passive smokers were excluded from the reference category.

For never smokers, FL risk increased for those exposed indoors to more than two smokers during their childhood only, but there was no significant trend with increasing numbers of smokers (Table 3). No association was observed between FL risk and indoor passive smoking exposure during adulthood only, during both childhood and adulthood, or at social venues as an adult. Results for passive smoking exposure were null when stratified by control type (Appendix A).

We found no association between FL risk and alcohol intake in the 12 months prior to enrolment or the alcohol type consumed (Table 4). The results were unchanged when stratified by control type (Appendix A).

### 3.2. Case All-Cause Mortality

The median follow-up time was 83 (IQR 70–98) months. During follow-up, 49 (7.0%) cases died, of which 23 (46.9%) were classified as FL-related, 11 (22.4%) as non-FL related and 15 (30.6%) as unknown cause. 

Compared with never smoking, current smoking was associated with a higher risk of death (HR = 3.90, 95%CI 1.79–8.53; Table 5). We found no association between earlier smoking initiation, or years since quitting and all-cause mortality. For current smokers, we observed a 5-fold excess risk of death for smoking <20 cigarettes per day (HR = 5.10, 95%CI 2.10–12.37) and non-significantly elevated risk for smoking ≥20 cigarettes per day (HR = 3.11, 95%CI 0.91–10.59), based on a small number of exposed cases. The highest categories of smoking duration (>27 years; HR = 3.24, 95%CI 1.71–6.15) and lifetime cigarette exposure (≥20 pack-years; HR = 2.54, 95%CI 1.29–5.00) were associated with increased risk of death. There was no meaningful change in these associations when the reference group excluded those who reported passive smoking. With further adjustment for stage of disease and first-line treatment we observed similar excess risks of death (Appendix A).

We observed no association between risk of death and indoor passive smoking exposure (Table 6). Similarly, there was no evidence of an association between consumption of alcohol of any kind or frequency of alcohol intake in the 12 months prior to enrolment and all-cause mortality (data not shown).

### 3.3. Case FL-Specific Mortality

Current smoking was associated with a non-significantly higher risk of FL-related death (HR = 2.97, 95%CI 0.91–9.72) based on a small number of exposed cases (*n* = 5; Table 5). The point estimate for current smoking was attenuated (HR = 2.65, 95%CI 0.68–10.35) after adjustment for stage of disease and first-line treatment (Appendix A). No association was observed with the highest category of pack-years smoked. We found no association between consumption of alcohol of any kind or frequency of intake and FL-specific mortality (data not shown).

Results from imputed analyses were consistent with findings without imputation (data not shown). Findings were also similar when cases with composite FL/DLBCL or grade 3B histology were excluded (Appendix A).

## 4. Discussion

Using a population-based family case-control study design, we found consistent evidence of increased FL risk with history of personal smoking. Several smoking characteristics were associated with both increased FL risk and all-cause mortality, and there was an indication that current smoking may be associated with FL-specific mortality. For never smokers, passive smoking during childhood was also associated with elevated FL risk. We observed no consistent evidence of an association between alcohol intake 12 months prior to enrolment and FL risk, all-cause mortality or FL-specific mortality.

We found a positive association between FL risk and ever, former and current smoking, and the highest categories of smoking frequency, duration and pack-years. These findings broadly align with those from the International Lymphoma Epidemiology Consortium (InterLymph) pooled analysis of 19 population and hospital-based case-control studies [33]. The pooled analysis identified increased FL risk with ever, former and current smoking, and a positive trend with longer duration of smoking, but no trend with increasing smoking frequency or pack-years. In contrast, the findings from cohort studies are inconsistent. Diver et al. [34] in the Cancer Prevention Study II Nutrition Cohort reported significant increased risk with current smoking in women (RR = 2.13, 95%CI 1.20–3.77) but not men (RR = 0.52, 95%CI 0.19–1.48), while Kroll et al. [35] in the UK Million women study observed non-significant elevated risk for current smoking (RR = 1.08, 95%CI 0.97–1.20). In the US Kaiser Permanente Medical Care Program Cohort Study, former smoking and intensity of smoking were associated with increased FL risk (RR = 1.9, 95%CI 1.2–2.9 and RR = 2.2, 95%CI 1.2–4.2, respectively) [36], while current smoking was associated with increased FL risk in the Iowa Women’s Health Study (RR = 2.3, 95%CI 1.0–5.0) [37]. Other cohort studies [9,38] found no association between former or current smoking and FL risk, most based on small numbers of exposed cases. In agreement with the prospective California Teachers Study, our findings for personal smoking were consistently modestly strengthened when those with exposure to passive smoking were excluded from the referent group [9]. 

Considering passive smoking, we found increased FL risk with indoor exposure to more than two smokers during childhood only, but not with exposures during adulthood, or both childhood and adulthood. This is partially consistent with the California Teachers Study, where Lu et al. reported a relative risk of 1.38 (95%CI 0.69–2.76) for household childhood passive smoking, 1.58 (95%CI 0.76–3.31) for household adulthood passive smoking and 2.02 (95%CI 1.06–3.87) for both childhood and adulthood household passive smoking exposure [9]. Additionally, they found significant trends in risk with increasing total years and increasing intensity of passive exposure to tobacco smoke in household, workplace and social settings combined during childhood and adulthood, but no significant finding with intensity-years of exposure. The lack of association in adulthood in our study may be because people are more likely to have control over their passive smoking exposure as an adult than during childhood. Passive smoking prevalence is higher in childhood than adulthood, and passive smoking exposure in childhood is mostly within the home, with children having no or very limited control over their exposure [39]. It is also noteworthy that children are more susceptible to the carcinogenic effects of passive smoking than adults. For example, children are less able to detoxify nitrosamine from cigarette smoke than adults [40]. Alternatively, it is possible that participants in our study were less able to accurately describe their exposure in adult settings, resulting in bias towards the null. Variation in the timing and extent of smoking laws in California and Australia during participants’ adult years may be another factor. Californians were likely to have more exposure to passive smoking as an adult compared to Australians. In California, a statewide ban on smoking in workplaces and indoor public spaces became law in 1995 [41], and participants were enrolled into the CTS in the same year. In contrast, smoking was banned in workplaces and public settings in New South Wales [42] and Victoria [43] 10–16 years before our study enrollment.

We found higher risk of all-cause mortality with current smoking and higher intensity, longer duration and higher pack-years of smoking. This is mostly consistent with a meta-analysis of five case-control studies [10]. The meta-analysis reported excess all-cause mortality with higher number of cigarettes smoked per day, longer duration and higher pack-years of smoking, but no association and moderate heterogeneity for current smoking. Our findings suggested a positive association between FL-specific mortality and current smoking, consistent with a case-control study that observed positive associations with current smoking, recently quitting smoking, and increasing duration and pack-years of smoking [11]. No prior cohort studies have investigated the association between personal or passive smoking exposure and FL-specific mortality.

A plausible biological mechanism by which smoking could increase FL risk is via chromosomal translocation. t(14:18) translocation is the first genetic event in FL pathogenesis [44,45,46], leading to dysregulated expression of the anti-apoptotic protein BCL2. A clonal analysis of t(14;18) translocation in healthy individuals before FL diagnosis and their paired FL tumour samples showed that progression to FL occurred from t(14;18)-positive committed precursors [6]. Smoking is associated with a higher prevalence of t(14:18) translocation; a 4-fold increased frequency among smokers compared to non-smokers [7]. Hence, smoking could be an initiating exposure in the molecular pathogenesis of FL. Our finding of similar FL risk for current and former smokers appears to support this. 

The mutational signature of FL appears not to strongly support tobacco smoke as a “second hit” in the pathway to malignant transformation. The major carcinogens in tobacco smoke are aromatic hydrocarbons and formaldehyde [47]. Aromatic hydrocarbons can induce DNA damage leading to somatic mutations, including in the *TP**53* gene [48]. Although *TP53* mutations are relatively infrequent (5–6%) in FL [49,50,51], they have been associated with early FL progression and adverse prognosis [52,53]. The most frequent somatic mutations in FL are in histone modifying genes, specifically *KMT2D* (40–89%) [54,55], *CREBBP* (33–70%) [50,55,56] and the immunoglobulin gene *IGHV* (79%) [57]. Whole exome sequencing of B-cell lymphoma has shown substitutions with increased C > T and T > C mutations, proposed to be a consequence of base misrepair from DNA damage [58,59], but these mutations are not specific to exposure to tobacco smoke [60]. FL is characterised by increased C > T/G mutations attributable to overactive DNA editing by the *APOBEC* cytosine deaminases [61]. This DNA nucleotide mutation has been observed in tobacco-related cancers but also in other cancers with no established association with tobacco smoking [60]. On the other hand, FL is associated with germline genetic variation in HLA-DRB1 [62,63], and genetic variation in the HLA-DRB1-amino acid haplotype has been linked with increased risk of FL among smokers compared to non-smokers [64], suggesting an interaction between smoking and HLA-DRB1-associated antigen presentation in FL etiology.

There is no established mechanistic pathway through which smoking influences FL survival, however, individuals with p53 mutated-FL have shorter survival [49,50,65,66], and smokers may potentially be less able to tolerate optimal anti-lymphoma therapies [67].

We observed no association between FL risk and the recent consumption of alcohol or specific alcohol types, consistent with findings from previous cohort studies [68,69,70,71,72,73] and a case-control study [74]. The most recent systematic review and meta-analysis showed an inverse association with current alcohol intake in cohort studies, but no individual cohort study observed a significant inverse association [8]. The meta-analysis also found no association with current alcohol intake in case-control studies, and no evidence of an association with former alcohol intake. The null associations we observed with the consumption of specific alcohol types and FL risk is consistent with previous cohort studies [68,69,73]. In contrast, Chang 2010 et al., in the California Teachers Study [29], reported increased FL risk with former wine intake (RR = 2.08, 95%CI 1.09–3.99), but no association with current wine intake or consumption of beer or liquor. Findings from the two studies to examine lifetime alcohol consumption and FL risk, were null. Jayasekara et al., in the Melbourne Collaborative Cohort Study [28] found no association between 10 g per day increment in lifetime alcohol consumption and FL risk, while the InterLymph pooled analysis of case-control studies reported no trend with increasing lifetime alcohol consumption [33].

Consistent with a previous population-based case-control study [11], we found no association between recent consumption of alcohol of any type or the quantity of alcohol intake and risk of death after FL diagnosis. The population-based case-control studies that examined lifelong alcohol consumption and risk of death also reported no association with younger age of initiation, increasing intensity, longer duration or lifetime exposure [12,13,14]. Previously, no cohort studies have examined the association between alcohol intake and FL-specific mortality. 

This is the first study to use a population-based family case-control design to investigate risk factors for FL. Compared with prior traditional case-control studies [3,75], we had a robust control participation rate. Family members are generally more willing to participate as controls, thus reducing potential bias that may arise from non-participation [30,76]. The use of sibling controls also reduces confounding by unmeasured early life and genetic factors [77]. We maximised case representativeness by recruiting cases via cancer registries in jurisdictions where new cancer diagnoses are notified by statute. We comprehensively assessed smoking history including passive smoking. 

Our study has several limitations. Being a family-based study, there will be some correlation of exposures between cases and related controls (siblings) as they tend to have the same childhood environment and are likely to engage in similar lifestyle behaviours [78]. We accounted for correlation of exposure between related cases and controls in our analyses. In our sensitivity analyses by control type, we observed attenuation of point estimates in the models restricted to related cases and controls, showing the possible effect of correlation of exposure among siblings compared with unrelated controls. As is typical for case-control studies, not all those who were eligible agreed to participate and the non-participation may have biased our results. We did not receive demographic or histologic information on cases who declined to participate, thus we cannot confirm the representativeness of the final analytical case sample. The smoking and alcohol history of non-participants may differ to that of participants, leading to over- or under-estimation of the true association. Differential recall of smoking and alcohol intake may also have biased our results; compared with controls, cases are more likely to over-estimate their exposure [79]. We used lifetime calendars to aid recall and minimise the potential for recall bias. Consistent with most prior studies, we did not ascertain lifetime alcohol consumption, and the referent group for the alcohol analyses was non-drinkers 12 months prior to enrolment, a group that will include both lifetime abstainers and former drinkers. This classification may have introduced bias, as former drinkers may have stopped drinking due to poor health related to FL, thus attenuating our risk estimates towards the null [80]. We adjusted for receipt of first-line treatment in our survival analyses, but we did not collect detailed information on treatment, or dates of relapse or progression. Finally, not all cases had a sibling or spouse control and there were small numbers of cases and controls in some exposure categories, limiting the statistical power to detect an association.

## 5. Conclusions

Our findings are consistent with previous studies showing an association between smoking and increased FL incidence and all-cause mortality after FL diagnosis, and no association with recent alcohol consumption. Our novel findings include an association between passive smoking as a child and increased FL risk, and a signal that smoking may increase FL-specific mortality. The totality of these epidemiological findings implicates tobacco smoke as carcinogenic for FL and possibly also the progression of this malignancy. They strengthen the evidence for ongoing multi-faceted tobacco control activities to reduce FL incidence, and to improve patient outcomes in newly diagnosed individuals. 

## Figures and Tables

**Table 1 cancers-14-02710-t001:** Characteristics of follicular lymphoma cases and controls.

Characteristics	Cases *n* (%)	Controls	
		Related *n* (%)	Unrelated *n* (%)
Total	709 (59.13)	303 (25.27)	187 (15.60)
Sex			
Male	368 (51.90)	123 (40.59)	77 (41.18)
Female	341 (48.10)	180 (59.41)	110 (58.82)
Twin status			
Twins	23 (3.24)	16 (5.28)	-
Identical (monozygotic)	11 (1.55)	9 (2.97)	-
Non-identical (dizygotic)	12 (1.69)	7 (2.31)	-
Non twin	674 (95.07)	283 (94.40)	187 (100.00)
Missing	12 (1.69)	4 (1.32)	-
Ethnicity			
Caucasian/white	664 (93.65)	288 (95.05)	171 (91.44)
Other	19 (2.68)	8 (2.64)	6 (3.21)
Missing	26 (3.67)	7 (2.31)	10 (5.35)
Stage at diagnosis ^a^			
I–II	181 (25.53)		
III–IV	349 (49.22)		
Missing	179 (25.25)		
Histologic grade at diagnosis ^a^			
1–2	488 (68.82)		
3A–3B ^b^	194 (27.36)		
Missing	27 (3.80)		
Composite FL/DLBCL ^c^	47 (6.63)		
FLIPI score at diagnosis ^a^			
Low (0–1)	179 (25.25)		
Intermediate (2)	123 (17.35)		
High (3–4)	140 (19.75)		
Missing	267 (37.66)		
First-line treatment ^a^			
None	166 (23.41)		
Chemotherapy	292 (41.18)		
Radiotherapy	46 (6.49)		
Chemotherapy/radiotherapy	31 (4.37)		
Missing	174 (24.54)		

^a^ Cases only; ^b^ Grade 3B = 44 cases; ^c^ FL/DLBCL = Follicular lymphoma and diffuse large B-cell lymphoma; n = number; FLIPI = Follicular Lymphoma International Prognostic Index.

**Table 2 cancers-14-02710-t002:** Odds ratios and 95% confidence intervals for FL risk in relation to personal smoking.

Exposures	Cases	Reference Category Included Passive Smokers				Reference Category Excluded Passive Smokers			
		Related Controls	Unrelated Controls	OR (95% CI) ^a^	*p*	Related Controls	Unrelated Controls	OR (95% CI) ^a^	*p*
Smoking status ^b^									
Never	369	175	118	Ref.	0.01	68	35	Ref.	0.01
Ever	340	127	69	1.38 (1.08–1.74)		127	69	1.51 (1.09–2.10)	
Smoking status ^b^									
Never	369	175	118	Ref.	0.03	68	35	Ref.	0.04
Former	274	106	56	1.36 (1.05–1.77)		106	56	1.50 (1.05–2.14)	
Current	66	21	13	1.43 (0.92–2.20)		21	13	1.56 (0.96–2.53)	
Age started smoking ^b^									
Never	369	175	118	Ref.	0.06	68	35	Ref.	0.08
Tertile 1 (>18)	97	29	19	1.47 (0.99–2.17)		29	19	1.62 (1.02–2.55)	
Tertile 2 (17–18)	110	54	25	1.20 (0.85–1.70)		54	25	1.34 (0.88–2.02)	
Tertile 3 (<17)	132	44	25	1.47 (1.06–2.05)		44	25	1.62 (1.08–2.41)	
				*P*_trend_ 0.17				*P*_trend_ 0.07	
Years since quitting smoking ^b^									
Never	369	175	118	Ref.	0.13	68	35	Ref.	0.19
Tertile 1 (≥30)	95	33	22	1.49 (0.99–2.24)		33	22	1.57 (0.97–2.53)	
Tertile 2 (15–29)	99	34	21	1.37 (0.94–1.99)		34	21	1.49 (0.95–2.35)	
Tertile 3 (<15)	80	39	13	1.24 (0.85–1.81)		39	13	1.41 (0.91–2.19)	
				*P*_trend_ 0.02				*P*_trend_ 0.05	
No. of cigarettes per day ^b^									
Never	369	175	118	Ref.	0.05	68	35	Ref.	0.04
<10	90	28	17	1.54 (1.02–2.32)		28	17	1.75 (1.10–2.77)	
10–19	105	49	24	1.19 (0.84–1.68)		49	24	1.32 (0.87–1.99)	
≥20	132	45	25	1.44 (1.04–2.01)		45	25	1.59 (1.07–2.38)	
				*P*_trend_ 0.06				*P*_trend_ 0.08	
Duration of cigarette smoking (years) ^b^									
Never	369	175	118	Ref.	0.04	68	35	Ref.	0.07
Tertile 1 (≤13)	114	45	28	1.24 (0.87–1.76)		45	28	1.35 (0.89–2.06)	
Tertile 2 (14–27)	106	49	16	1.42 (0.99–2.03)		49	16	1.60 (1.05–2.44)	
Tertile 3 (>27)	119	33	25	1.53 (1.07–2.18)		33	25	1.64 (1.07–2.51)	
				*P*_trend_ < 0.01				*P*_trend_ 0.01	
Lifetime cigarette exposure (pack-years) ^b^									
Never	369	175	118	Ref.	0.05	68	35	Ref.	0.23
Tertile 1 (<6.8)	111	40	21	1.43 (0.98–2.06)		40	21	1.60 (1.05–2.46)	
Tertile 2 (6.8–19.9)	102	49	22	1.16 (0.81–1.67)		49	22	1.33 (0.87–2.05)	
Tertile 3 (≥20.0)	113	33	23	1.56 (1.10–2.22)		33	23	1.66 (1.09–2.53)	
				*P*_trend_ 0.02				*P*_trend_ 0.06	

^a^ Multivariable model—adjusted for age, sex, ethnicity, state, quantity of alcohol intake 12 months prior to enrolment. ORs are based on related and unrelated controls combined. ^b^ Imputations (number of participants with missing values): ever smoking status (1), age started smoking (1), year since quit smoking (1), number of cigarettes per day (22), duration of smoking (2), pack-years (23).

**Table 3 cancers-14-02710-t003:** Odds ratios and 95% confidence intervals for FL risk in relation to passive smoking exposure among never smokers.

Passive Smoking	Cases	Related Controls	Unrelated Controls	OR (95% CI) ^a^	*p*
Never smokers with no passive smoking exposure	109	68	35	Ref.	
Childhood only passive smoking ^b^					
Intensity (no. of smokers) ^b^					
1	109	38	37	1.22 (0.82–1.82)	0.05
2	46	28	19	0.85 (0.52–1.38)	
>2	67	21	13	1.84 (1.11–3.04)	
				*P*_trend_ 0.09	
Duration (years) ^b^					
1–6	72	25	26	1.23 (0.78–1.95)	0.62
7–10	67	31	17	1.23 (0.78–1.95)	
>10	73	31	23	1.18 (0.76–1.84)	
Adulthood only passive smoking ^b^					
Intensity (no. of smokers) ^b^					
1	41	23	17	0.94 (0.52–1.68)	0.42
2–4	55	31	15	1.01 (0.60–1.69)	
>4	65	21	18	1.44 (0.87–2.39)	
				*P*_trend_ 0.22	
Duration (years) ^b^					
≤6	35	19	17	0.88 (0.50–1.56)	0.39
7–18	60	27	16	1.21 (0.74–1.99)	
>18	58	28	14	1.32 (0.78–2.27)	
				*P*_trend_ 0.21	
Childhood and adulthood passive smoking ^b^	259	108	83	1.20 (0.85–1.68)	0.30
Social venue passive smoking as an adult ^b^					
Duration (years) ^b^					
≤2	56	20	17	1.27 (0.74–2.17)	0.57
>2	29	19	4	1.00 (0.48–2.09)	

^a^ Multivariable model—adjusted for age, sex, ethnicity, state, quantity of alcohol intake 12 months prior to enrolment. ORs are based on related and unrelated controls combined. ^b^ Imputations (number of participants with missing values): childhood passive smoking—intensity (11), duration (24); adulthood—intensity (8), duration (20); childhood or adulthood (6); social venues—duration (11).

**Table 4 cancers-14-02710-t004:** Odds ratios and 95% confidence intervals for FL risk in relation to alcohol intake 12 months prior to enrolment.

Exposures	Cases	Related Controls	Unrelated Controls	OR (95% CI) ^a^	*p*
Alcohol intake					
No	79	31	24	Ref.	0.10
Yes	630	272	163	1.00 (0.69–1.46)	
Frequency of any alcohol intake (per week)					
None	79	31	24	Ref.	0.45
<once	183	82	44	1.09 (0.71–1.68)	
once	66	21	17	1.23 (0.71–2.15)	
>once	381	169	102	0.93 (0.63–1.36)	
Quantity of any alcohol intake (grams of ethanol/day) ^b^					
None	79	31	24	Ref.	0.18
>5.20	210	94	58	1.10 (0.72–1.69)	
5.20–19.70	221	80	58	1.09 (0.71–1.67)	
>19.70	198	98	46	0.81 (0.54–1.23)	
				*P*_trend_ 0.14	
Beer intake					
No	232	125	75	Ref.	0.82
Yes	398	147	88	1.04 (0.74–1.45)	
Frequency of beer intake (per week)					
None	232	125	75	Ref.	0.95
<once	179	72	39	1.06 (0.74–1.51)	
once	55	18	16	0.92 (0.52–1.59)	
>once	164	57	33	1.05 (0.69–1.61)	
Quantity of beer intake (grams of ethanol/day) ^b^					
None	232	125	75	Ref.	0.87
<1.46	150	61	33	1.08 (0.75–1.75)	
1.46–7.76	112	33	30	0.93 (0.59–1.44)	
>7.76	134	53	25	0.93 (0.58–1.47)	
				*P*_trend_ 0.66	
Quantity of beer that was light beer ^b^					
None	232	125	75	Ref.	0.45
Almost none	190	73	45	0.90 (0.62–1.32)	
Less than half	38	19	8	0.79 (0.43–1.45)	
About half	46	19	8	1.03 (0.57–1.86)	
More than half	17	9	3	0.70 (0.29–1.67)	
All or almost all	101	27	23	1.32 (0.85–2.05)	
Wine intake					
No	82	28	15	Ref.	0.53
Yes	548	244	148	0.87 (0.58–1.32)	
Frequency of wine intake (per week)					
None	82	28	15	Ref.	0.21
<once	191	79	44	0.99 (0.62–1.60)	
once	77	22	14	1.35 (0.76–2.39)	
>once	280	143	90	0.74 (0.48–1.12)	
Quantity of wine intake (grams of ethanol/day) ^b^					
None	82	28	15	Ref.	0.29
<2.98	224	84	55	1.09 (0.68–1.73)	
2.98–14.49	186	77	53	0.88 (0.57–1.38)	
>14.49	137	83	39	0.67 (0.42–1.06)	
				*P*_trend_ 0.27	
Quantity of wine that was red wine ^b^					
None	82	28	15	Ref.	0.78
Almost none	142	66	55	0.84 (0.52–1.34)	
Less than half	79	34	19	0.91 (0.54–1.55)	
About half	103	46	26	0.87 (0.53–1.43)	
More than half	53	31	12	0.70 (0.40–1.22)	
All or almost all	170	67	35	0.98 (0.61–1.56)	
Spirit intake					
No	274	121	70	Ref.	0.85
Yes	356	151	93	1.03 (0.79–1.33)	
Frequency of spirit intake (per week)					
None	274	121	70	Ref.	0.98
<once	263	119	67	1.02 (0.77–1.35)	
once	40	12	8	1.27 (0.71–2.26)	
>once	53	20	18	0.89 (0.55–1.44)	
Quantity of spirit intake (grams of ethanol/day)					
None	274	121	70	Ref.	0.97
<0.24	132	61	35	0.99 (0.71–1.38)	
0.24–1.23	113	48	26	1.04 (0.70–1.54)	
>1.23	111	42	32	0.93 (0.64–1.34)	

^a^ Multivariable model—adjusted for age, sex, ethnicity, state, smoking (never, current, former); estimates of beer, wine and spirits intake were mutually adjusted for each other. ORs are based on related and unrelated controls combined. ^b^ Imputations (number of participants with missing values): quantity of any alcohol (2), quantity of beer (2), light beer (7), quantity of wine (2) and red wine intake (2).

**Table 5 cancers-14-02710-t005:** Hazard ratios and 95% confidence intervals for all-cause mortality and FL-specific mortality in relation to smoking.

Exposures	No. of Deaths/Person-Months	Reference Category Included Passive Smoking		Reference Category Excluded Passive Smokers	
		HR (95% CI) ^a^	*p*	HR (95% CI) ^a^	*p*
All-cause mortality					
Smoking status					
Never	18/31,022	Ref.	0.10	Ref.	0.34
Ever	31/27,802	1.65 (0.91–2.98)		1.59 (0.61–4.15)	
Smoking status					
Never	18/31,022	Ref.	<0.01	Ref.	0.01
Former	20/22,572	1.27 (0.67–2.41)		1.20 (0.45–3.18)	
Current	11/5230	3.90 (1.79–8.53)		3.69 (1.26–10.82)	
Age started smoking (years)					
Never	18/31,022	Ref.	0.23	Ref.	0.53
≥18	16/14,022	1.61 (0.79–3.27)		1.55 (0.55–4.33)	
<18	15/13,698	1.71 (0.87–3.37)		1.66 (0.60–4.59)	
Years since quitting smoking					
Never	18/31,022	Ref.	0.24	Ref.	0.24
≥20	10/13,883	0.98 (0.45–2.16)		0.98 (0.45–2.16)	
<20	10/8689	1.87 (0.86–4.06)		1.87 (0.86–4.06)	
No. of cigarettes per day ^b^					
Never~~~Former smokers	18/31,022	Ref.		Ref.	
<20	9/13,103	1.02 (0.46–2.28)	0.73	0.97 (0.32–2.94)	0.91
≥20	10/8860	1.55 (0.71–3.39)		1.45 (0.59–4.32)	
Current smokers					
<20	7/3073	5.10 (2.10–12.37)	<0.01	4.78 (1.51–15.12)	0.02
≥20	4/1766	3.11 (0.91–10.59)		2.95 (0.70–12.39)	
Duration of cigarette smoking (years)					
Never	18/31,022	Ref.	<0.01	Ref.	<0.01
Tertile 1 (≤13)	4/9507	0.55 (0.16–1.87)		0.54 (0.13–2.27)	
Tertile 2 (14–27)	5/8783	0.91 (0.34–2.45)		0.91 (0.26–3.17)	
Tertile 3 (>27)	22/9430	3.24 (1.71–6.15)		3.25 (1.20–8.83)	
		*P*_trend_ < 0.01		*P*_trend_ 0.01	
Lifetime cigarette exposure (pack-years) ^b^					
Never	18/31,022	Ref.	0.04	Ref.	0.20
Tertile 1 (<6.8)	5/9005	0.80 (0.28–2.35)		0.82 (0.22–3.00)	
Tertile 2 (6.8–19.9)	8/8628	1.50 (0.65–3.47)		1.50 (0.48–4.65)	
Tertile 3 (≥20.0)	17/9088	2.54 (1.29–5.00)		2.51 (0.90–7.00)	
		*P*_trend_ 0.01		*P*_trend_ 0.10	
FL-specific mortality					
Smoking status					
Never	9/31,022	Ref.	0.41	Ref.	0.45
Ever	14/27,802	1.43 (0.61–3.39)		1.77 (0.39–7.95)	
Smoking status					
Never	9/31,022	Ref.	0.18	Ref.	0.18
Former	9/22,572	1.16 (0.45–2.95)		1.16 (0.45–2.95)	
Current	5/5230	2.97 (0.91–9.72)		2.97 (0.91–9.72)	
Lifetime cigarette exposure (pack-years) ^b^					
Never	9/31,022	Ref.	0.77	Ref.	0.46
<20	7/17,633	1.09 (0.39–3.08)		1.14 (0.40–3.21)	
≥20	6/9088	1.67 (0.59–4.77)		1.87 (0.68–5.13)	

^a^ Basic model—adjusted for age, sex, ethnicity, state. ^b^ Imputations (number of participants with missing values): age started smoking (1), no. of cigarettes per day (13), duration (1), pack-years (14).

**Table 6 cancers-14-02710-t006:** Hazard ratios and 95% confidence intervals for all-cause mortality after FL diagnosis in relation to passive smoking exposure among never smokers.

Passive Smoking	Person-Months	No. of Deaths	All-Cause Mortality	
			HR (95% CI) ^a^	*p*
Never smokers with no passive smoking exposure	9055	5	Ref.	
Childhood only passive smoking				
Intensity (no. of smokers)				
<2	9517	6	0.97 (0.29–3.28)	0.97
≥2	9475	5	0.87 (0.25–3.06)	
Duration (years)				
<7	6029	4	0.97 (0.26–3.71)	0.19
≥7	11,982	7	0.97 (0.30–3.10)	
Adulthood only passive smoking				
Intensity (no. of smokers)				
≤4	8056	6	1.24 (0.37–4.17)	0.84
>4	5470	5	1.47 (0.41–5.22)	
Duration (years) ^b^				
≤18	7982	6	1.37 (0.42–4.39)	0.79
>18	4964	4	1.34 (0.34–5.20)	
Childhood and adulthood passive smoking	21,875	13	0.95 (0.33–2.70)	0.92
Social venues passive smoking	12,045	7	0.93 (0.35–2.46)	0.89

^a^ Basic model—adjusted for age, sex, ethnicity, state. ^b^ Imputation (number of participants with missing values): adulthood passive smoking—duration (1).

## Data Availability

The data that support the findings of our study originate from the Lymphoma, Lifestyle, Environment and Family (LEAF) Study, and cause of death data were provided by the Australian Institute of Health and Welfare. Data can be made available upon request.

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
