# Peer review of "Associations between Smoking and Alcohol and Follicular Lymphoma Incidence and Survival: A Family-Based Case-Control Study in Australia"

_cancers, 2022, doi:10.3390/cancers14112710_

Round 1
Reviewer 1 Report
Very interesting article and a complete and precise study. Few suggestions and points to consider:
Previous studies show a relation between HLA-DRB1 and risk of FL, did you consider such relation and it's effect on your study?
Some studies show relation between smoking and immunodeficiency and recommend joint assessment of smoking parameters and biomarkers of infectious agents, specially for EBV infection, did you consider checking it in your study population?
Any other confounding factor other than alcohol, including age, obesity, SES, etc ?
Did you consider the possibility of recall bias (information bias), in your population specially passive smokers?
Reviewer 2 Report
The manuscript submitted by Odutola et al. describes an epidemiological case-control study trying to decipher the links between smoking history – including passive exposure- and alcohol consumption with Follicular Lymphoma (FL) risk and mortality. Although well designed and presented, this study does not convey a particularly new or original message. Increased OR are small or on the verge of significance, or even lacking.
Some discrepancies in the results (either in the present study or with previous published data) are not discussed in enough details.
Finally, some biological or functional experiments, which could bring some mechanistic explanations, would greatly enhance the quality of this study.
Comments:
- lines 162-165: If childhood is defined as age < 18 yrs, how can exposure be more than 20 yrs?
- Discussion: it would be more suitable to compare OR between the current work and cohort studies, rather than OR in one case and RR in others.
- Discussion: association with passive smoking during childhood only, but not childhood and adulthood is this study compared to associated to both in other cited studies should be further discussed. What could be the causes of this discrepancy? It seems counter-intuitive to have association only with passive smoking during childhood and not during both time periods.
- Discussion: association of FL risk and smoking due to increased t(11;14) induced by smoking may also be related to duration / intensity of smoking. A longer / higher exposure should increase the risk, which is not really visible in this study.
Is it possible to make a comparison of the frequency t(11;14) in smokers / non-smokers (possibly before and after FL diagnosis) or to bring any biological data that sustain this discussed point?
Round 2
Reviewer 2 Report
The authors answered to all my comments, and i would like to thank them for their answers.
I do not have any further comments, and therefore recommend this manuscript for publication in Cancers.